# Salivary Tryptophan as a Metabolic Marker of HER2-Negative Molecular Subtypes of Breast Cancer

**DOI:** 10.3390/metabo14050247

**Published:** 2024-04-25

**Authors:** Elena A. Sarf, Elena I. Dyachenko, Lyudmila V. Bel’skaya

**Affiliations:** Biochemistry Research Laboratory, Omsk State Pedagogical University, 644099 Omsk, Russia; sarf_ea@omgpu.ru (E.A.S.); dyachenko.ea@gkpc.buzoo.ru (E.I.D.)

**Keywords:** saliva, capillary electrophoresis, amino acids, tryptophan, breast cancer, HER2, Ki-67, progesterone

## Abstract

Changes in the concentration of tryptophan (Trp) indicate a serious metabolic restructuring, which is both a cause and a consequence of many diseases. This work examines the upward change in salivary Trp concentrations among patients with breast cancer. This study involved volunteers divided into three groups: breast cancer (n = 104), non-malignant breast pathologies (n = 30) and healthy controls (n = 20). In all participants, before treatment, the quantitative content of Trp in saliva was determined by capillary electrophoresis. In 20 patients with breast cancer, Trp was re-tested four weeks after surgical removal of the tumor. An increase in the Trp content in saliva in breast cancer has been shown, which statistically significantly decreases after surgical removal of the tumor. A direct correlation was found between increased Trp levels with the degree of malignancy and aggressive molecular subtypes of breast cancer, namely triple negative and luminal B-like HER2-negative. These conclusions were based on an increase in Ki-67 and an increase in Trp in HER2-negative and progesterone-negative subtypes. Factors under which an increase in Trp concentration in saliva was observed were identified: advanced stage of breast cancer, the presence of regional metastasis, low tumor differentiation, a lack of expression of HER2, estrogen and progesterone receptors and the high proliferative activity of the tumor. Thus, the determination of salivary Trp may be a valuable tool in the study of metabolic changes associated with cancer, particularly breast cancer.

## 1. Introduction

Breast cancer is the most common female cancer pathology worldwide [1]. Breast cancer is gradually leading in the structure of deaths from cancer, despite the fact that with early detection, the 5-year survival rate reaches 90% [2]. In this regard, the search for new methods for diagnosing breast cancer does not lose its relevance. Work in this direction is complicated by the fact that breast cancer is a heterogeneous disease and, often, the same biomarker for different molecular biological subtypes of breast cancer can show dramatically different values [3]. This makes it necessary to consider molecular biological subtype when analyzing the potential diagnostic performance of breast cancer biomarkers.

It is known that the development and progression of breast cancer is inextricably linked with glucose and lipid metabolism [4]. This process generates energy and amino acids for metabolic homeostasis. It should be noted that amino acids are essential nutrients for all living cells, while tumor cells have a higher requirement for amino acids compared to normal cells [5]. It has been found that in breast cancer, amino acid metabolism is reprogrammed [5]. One of these amino acids is tryptophan (Trp), which affects cell proliferation and can influence the survival of immune cells in the tumor microenvironment [6]. The Trp metabolism is known to be associated with breast cancer progression [7], immune response [8], and choice of therapeutic strategy [9,10].

In recent years, interest in saliva as a diagnostic tool has increased [11,12], as saliva can be collected non-invasively and repeatedly without the discomfort associated with blood sampling [13]. Saliva is already widely used in genetic testing due to its better transport stability compared to blood [14]. Saliva contains various substances and biomarkers that can be used as indicators of health and disease, including for the diagnosis of cancer [15,16,17]. Particular attention is paid to the analysis of salivary amino acids in oncological diseases, including breast cancer [18,19]. However, the content of Trp in saliva in breast cancer has not been considered separately until now, despite its mention in the list of amino acids that were analyzed for this pathology [20,21,22,23].

We previously developed a method for the quantitative determination of Trp in saliva by capillary electrophoresis and showed that the content of Trp in saliva in breast cancer patients is increased compared to healthy controls [24]. In this study, we analyzed the quantitative content of Trp in the saliva of patients with breast cancer, non-malignant breast pathologies and healthy controls, and assessed the relationship between the level of salivary Trp and the molecular characteristics of the tumor.

## 2. Materials and Methods

### 2.1. Description of the Study Group

The study included 133 patients from the Omsk Clinical Oncology Center. Patients were included after informed consent; the study was conducted after approval by the ethics committee of the Omsk Clinical Oncology Center (21 July 2016, protocol No. 15) and in accordance with the Helsinki principles.

The case–control study involved volunteers who were divided into 3 groups: the main group (breast cancer, n = 103, age 53.3 ± 3.3 years), the comparison group (fibroadenomas, n = 30, age 48.9 ± 4.3 years) and a control group (healthy control, n = 20, age 45.9 ± 7.1 years).

Inclusion in groups occurred in parallel. The inclusion criteria were the following: female gender; patient aged 30–60 years; absence of any treatment at the time of the study, including surgery; chemotherapy or radiation; and absence of signs of active infection (including purulent processes). All participants were examined by a dentist and had good oral hygiene. Exclusion criteria: a lack of histological verification of the diagnosis. In the control group volunteers, routine mammography and ultrasound examination showed the absence of mammary gland pathologies.

### 2.2. Collection of Saliva Samples

Saliva samples were collected in the morning on an empty stomach by spitting into sterile polypropylene tubes after first rinsing the mouth with boiled water. Saliva samples from all patients were collected strictly before the start of treatment. In 20 patients with breast cancer, Trp was retested four weeks after surgical removal of the tumor before starting adjuvant chemotherapy.

Saliva samples were collected for 10–15 min without additional stimulation. For all subjects, the flow rate of saliva (mL/min) was calculated. The saliva flow rate for patients with breast cancer was 0.93 [0.62; 1.25] mL/min; for non-malignant pathologies of the mammary glands, it was 0.96 [0.63; 1.29] mL/min; and for healthy controls, it was 0.97 [0.90; 1.04] mL/min. There were no statistically significant differences in these parameters between the subgroups. The concentration of total protein in saliva (g/L) was determined by reaction with pyrogallol red. The presence of a correlation between protein concentration and Trp in saliva has not been confirmed (*r* = 0.0848).

### 2.3. Determination of Tryptophan in Saliva by Capillary Electrophoresis

A 200 μL saliva sample was mixed with 200 μL of 16% trichloroacetic acid to precipitate proteins and then centrifuged at 10,000× *g* for 10 min (CLb-16, Moscow, Russia), 100 μL of the supernatant liquid was taken and the volume was adjusted to 1 mL with double-distilled water. The final dilution was 20 times. The experiment was carried out using a KAPEL-105M capillary electrophoresis system with a positive-polarity high-voltage source (Lumex, St. Petersburg, Russia). The leading electrolyte is sodium tetraborate 0.02 mol/L. To carry out the research, a quartz capillary was used: effective length—65 cm, total length—75 cm, and internal diameter—50 µm. The analysis conditions were as follows: sample injection into the pneumatic capillary—30 mbar, sample injection time—5 s, constant voltage—25 kV, temperature—30 °C, analysis time—4–5 min, and working wavelength of the photometric detector—219 nm.

Quantitative determination of Trp was carried out using a previously constructed calibration graph y = 0.8483x (r^2^ = 0.9998). Trp solutions with concentrations of 10, 15 and 20 μg/L were used to plot the graph. The accuracy and reproducibility of Trp determination in saliva samples was confirmed by spiking with a known Trp concentration.

### 2.4. Determination of the Expression of the Receptors for Estrogen, Progesterone, HER2 and Ki-67

The Allred Scoring Guideline was used to assess the level of expression of estrogen receptors (ERs), progesterone receptors (PRs), and HER2 [25]. The level of expression of estrogen and progesterone receptors and HER2 was assigned to one of four categories (−, +, ++, +++) in accordance with the ASCO/CAP recommendations [26]. Ki-67 expression was determined as part of a standard breast cancer panel according to the manufacturer’s protocol [27]. The cut-off value for Ki-67 was defined as 14% (low Ki-67) and 40% (high Ki-67). According to the obtained values, breast cancer was classified into five groups: triple negative breast cancer (TNBC), luminal A-like, luminal B-like (HER2-negative), luminal B-like (HER2-positive), and HER2-enriched (non-luminal).

### 2.5. Statistical Analysis

Statistical analysis of the obtained data was performed in the Statistica 13.3 EN software (StatSoft, Tulsa, OK, USA) using the nonparametric Mann–Whitney U test method. The sample was described using the median (Me) and interquartile range in the form of the 25th and 75th percentiles [LQ; UQ]. Differences were considered statistically significant at *p* ˂ 0.05.

## 3. Results

The Trp content in the saliva of healthy controls was 46.40 [44.79; 53.20] µg/L, whereas in non-malignant pathologies of the mammary glands (fibroadenomas), the Trp content increased slightly to 59.19 [43.85; 82.01] µg/L (*p* = 0.0960). In breast cancer, the Trp content increased significantly to 90.98 [62.68; 146.1] µg/L compared with both the control group (*p* < 0.0001) and fibroadenomas (*p* = 0.0039). Below in the text, p-values are given in comparison with fibroadenomas; the differences with the control group are statistically significant in all cases (*p* < 0.05).

The Trp content in saliva significantly depended on the stage of breast cancer, its molecular biological subtype, the status of lymph node involvement, etc. (Table 1).

It was shown that with increasing stage, according to the TNM system, the Trp content in saliva increased by 50.6% when comparing stages I and IV (Figure 1A), while an increase in Trp content in saliva by 26.0% was also noted when regional lymph nodes were affected (Figure 1D).

Of interest was the higher Trp content in saliva for TNBC and luminal B HER2-negative molecular biological breast cancer subtypes (Figure 1B). The differences between the Trp content in saliva for these breast cancer subtypes compared to others were statistically significant (TNBC vs. luminal A-like: +67.8%, *p* = 0.0149; TNBC vs. luminal B-like (HER2+): +61.1%, *p* = 0.0027; TNBC vs. non-luminal: +126.0%, *p* < 0.0001; luminal B-like (HER2-) vs. non-luminal: +61.5%, *p* = 0.0456). For the remaining molecular biological subtypes of breast cancer, the increase in Trp content in saliva was not statistically significant (*p* > 0.05).

There was no change in salivary Trp concentration depending on estrogen receptor status, whereas with the negative expression of progesterone receptors, Trp levels increased by an average of 14.1% (Figure 1C). However, this increase was not statistically significant (*p* = 0.5309).

Additionally, we determined the Trp content depending on HER2 expression (Figure 1F). It was found that in HER2-negative status, the Trp content increases by an average of 35.9% compared to HER2-positive status (*p* = 0.0027). The Trp level increases in HER2-negative and Ki-67 > 40% (Figure 1G), which corresponds to the molecular biological subtypes of TNBC (=HER2(−), ER(−), PR(−)) and luminal B-like HER2-negative subtype (=HER2(−), ER(+++), or PR low, or Ki-67 high). These same subtypes most often include low-grade cancers, which also increase salivary Trp levels (Figure 1E). Thus, out of 39 patients with GIII, 12 had TNBC (30.7%), and 8 had luminal B-like HER2-negative (20.5%).

Of the 103 patients, we randomly selected 20 to have a repeat Trp level measured 4 weeks after surgery before starting the first course of adjuvant chemotherapy. We compared the content of Trp in saliva of patients before and after surgery. It was shown that the Trp content decreased to 39.91 [31.77; 52.20] µg/L, which is even slightly lower than the corresponding values for the control group (Figure 2). However, no statistically significant differences with the control group were found.

The maximum decrease in Trp was noted for non-luminal breast cancer (31.30 [29.82; 33.78] μg/L). There were no differences in Trp content after surgery among all other molecular biological subtypes; the average Trp content in saliva was 50.39 [41.74; 59.63] µg/L.

## 4. Discussion

It is known that Trp is converted in the body through three key pathways: serotonin, indole, and kynurenine [28]. The kynurenine metabolic pathway is intensified in tumor tissue, which is explained by the activating effect of the c-MYC proto-oncogene, as well as genes overexpressed in tumors and responsible for the enzymes of kynurenine production: tryptophan-2,3-dioxygenase, indoleamine-2,3-dioxygenase and arylformamidase [29]. Upregulation of indoleamine 2,3-dioxygenase expression results in increased Trp metabolism, which increases the production of kynurenine, which in turn suppresses T-cell differentiation and consequently promotes cancer growth and development [30,31].

A number of studies have determined the Trp content in saliva in breast cancer. Thus, Sugimoto et al. [20] and Cheng et al. [21] showed that the Trp content in saliva in breast cancer increases approximately 1.5 times. Cheng et al. showed that the increase in Trp concentration in saliva is more pronounced for stages I–II breast cancer than for stages III–IV (2.07 and 1.56 times, respectively) [21]. Apparently, the higher Trp concentration in the early stages of breast cancer is due to the fact that as the cancer progresses, there is an increase in Trp catabolism, which can lead to a decrease in its concentration and the accumulation of kynurenine metabolites [32]. The concentration of Trp in the saliva of healthy volunteers was 49.3 ± 64.5 ng/mL, while in breast cancer, it was 101.9 ± 80.4 ng/mL for early stages; the differences between the groups were statistically insignificant (*p* = 0.497) [21]. In Sugimoto’s work, concentrations were not given; only a relative increase in Trp content by 1.59 times was shown in breast cancer compared with the control group [20]. In other types of cancer, the Trp content in saliva was also increased, in particular in oral squamous cell carcinoma [33] and thyroid cancer [34]. In the first case, a change in Trp concentration was shown compared to the control group (1.9 times); in the second, a threshold value was indicated, above which thyroid cancer can be diagnosed with an accuracy of 73.2% (53.7 ng/mL). Cheng et al. also showed that early and advanced breast cancer can be diagnosed with an accuracy of 76.3 and 78.6% at a threshold Trp concentration in saliva of 46.1 and 45.1 ng/mL, respectively [21]. The Trp concentration values we obtained correspond to the literature data and demonstrate an increase of 1.53 times in breast cancer compared to fibroadenomas and 2.11 times compared to the control group. The high heterogeneity of breast cancer and the wide range of values depending on the molecular biological characteristics of the tumor suggest that higher sensitivity values can be obtained for individual breast cancer subtypes with larger sample sizes. It should be noted that the Trp content in saliva in breast cancer was determined only in a few studies, while most indicated the ratio of concentrations in cancer and normal, without indicating absolute concentrations and the range in variation [20,21]. In those works, where average values are given, attention is drawn to a fairly wide spread of values, which may be due to the small sample size of 30/87 [20] and 27/28 [21] people in the main and control groups, respectively, as well as its heterogeneity in terms of race and age.

An interesting result of our work was the identification of a significant increase in the concentration of Trp in saliva in TNBC. It was previously shown that it is Trp metabolites in the blood plasma that make it possible to differentiate the TNBC group from other BC subtypes [35,36]. In particular, the content of N-acetyl-D-tryptophan allows, with a sensitivity of 90% and a specificity of 91.3%, to separate the TNBC and non-TNBC groups from each other, as well as the group of breast cancer patients and healthy controls with a sensitivity of 71.9% and a specificity of 92.5% [37]. At the same time, the content of Trp metabolites decreases in general in breast cancer compared to controls, but increases in TNBC [37]. A study by Li et al. showed that kynurenine levels increased in patients with TNBC compared with patients without TNBC, and its levels increased in the ER-negative and PR-negative groups, but decreased in the HER2-negative group [38]. Thus, the Trp content is apparently regulated by receptors that determine the classification of breast cancer into one or another molecular biological subtype.

In particular, we have shown that there is an increase in the concentration of Trp in saliva with a negative status of progesterone receptors and high expression values of the proliferation marker Ki-67, which apparently determines high concentrations of Trp in the subgroup of luminal B HER2-negative breast cancer. This fact is new and has not previously been mentioned in literary sources.

It is worth considering the increase in Trp as follows. When analyzing the data presented in Table 1, it becomes apparent that increased salivary Trp is directly associated with decreased cancer cell differentiation and with aggressive subtypes of breast cancer. This is supported by an increase in Ki-67 in samples with high Trp levels. This amino acid is also increased in the HER2(−) subtype of breast cancer, as well as in progesterone-negative and estrogen-negative subtypes of breast cancer. The hormone negative subtype of breast cancer indicates a high degree of malignancy of the tumor, since the absence of hormonal receptors reflects a low degree of differentiation of the cancer cell [39,40]. Trp in 90% of cases is metabolized through the kynurenine pathway, in which the accumulation of intermediate metabolites occurs, triggering oxidative stress and an altering effect on healthy cells, thereby constantly stimulating and maintaining states of inflammation and stress [4]. The direction of Trp metabolism along the kynurenine pathway occurs due to the activation of enzymes such as IDO and TDO, which, in turn, are activated by exposure to IL-1β, IL-6, α-TNF and cortisol, respectively (Figure 3) [41,42,43,44]. Thus, a “vicious cycle” is terminated: Neoplasm occurs due to many factors, including stress and the activation of inflammatory factors, which trigger the metabolic cascade described above. In turn, the presence of oncology itself is a trigger for the launch of the same metabolic pathway. A number of studies have shown that progesterone can have an inhibitory effect on IDO and TDO [45,46,47,48]; as a result, when the expression of progesterone receptors decreases, Trp metabolism is triggered along the kynurenine pathway [49,50].

It is known that amino acids from blood plasma enter the cells of the salivary glands through sodium-dependent active membrane transport systems [51]. They are predominantly involved in the process of synthesis of salivary proteins. Metabolism in the salivary gland itself may also play an important role in differences in salivary and blood profiles [37]. To understand the reason for the differences in salivary amino acid profiles, further validation of these results by comparing saliva profiles with blood and tissue profiles is necessary. We have previously shown that the content of amino acids, including Trp, in the blood plasma according to data from different authors varies significantly; however, on average, a decrease in the concentration of Trp is observed in breast cancer compared to the control group [18]. Research into the reasons for differences in Trp concentrations in saliva and serum in breast cancer has not yet been conducted. It can be hypothesized that in saliva, with increasing stages of breast cancer and with its aggressive cell subtypes, an increase in Trp reflects an increased need for this amino acid from the point of view of the systemic nature of metabolism at the level of digestion. A decrease in the concentration of the same amino acid in the blood serum with a parallel accumulation of its metabolites may be the result of the active consumption of Trp at the local level by the cancer cell itself to maintain its vital activity. In this case, we register the very fact of Trp metabolism in the body. Apparently, a separate substantiation is required for the identified fact of an increase in Trp concentration in saliva in breast cancer, which is planned to be performed in the next stages of the study.

Other limitations include the small sample size of patients with breast cancer and the control group, the determination of only Trp without kynurenine metabolites, and the lack of data on concomitant diseases, dietary patterns and the level of protein catabolism. In future, it is necessary to check whether the identified patterns are associated with breast cancer or whether they are characteristic of cancer diseases in general.

## 5. Conclusions

Thus, we identified factors under which an increase in Trp concentration in saliva was observed: advanced stage of breast cancer, the presence of metastasis in the lymph nodes, low tumor differentiation, lack of expression of HER2, estrogen and progesterone receptors, and a high proliferative activity of the tumor. The relationship between elevated Trp levels in saliva and breast cancer was confirmed by a decrease in Trp levels after surgical resection of the tumor. Although much remains to be studied and explained, it has now been shown that the determination of salivary Trp can be a valuable tool in the study of metabolic changes associated with cancer, particularly breast cancer.

## Figures and Tables

**Figure 1 metabolites-14-00247-f001:**
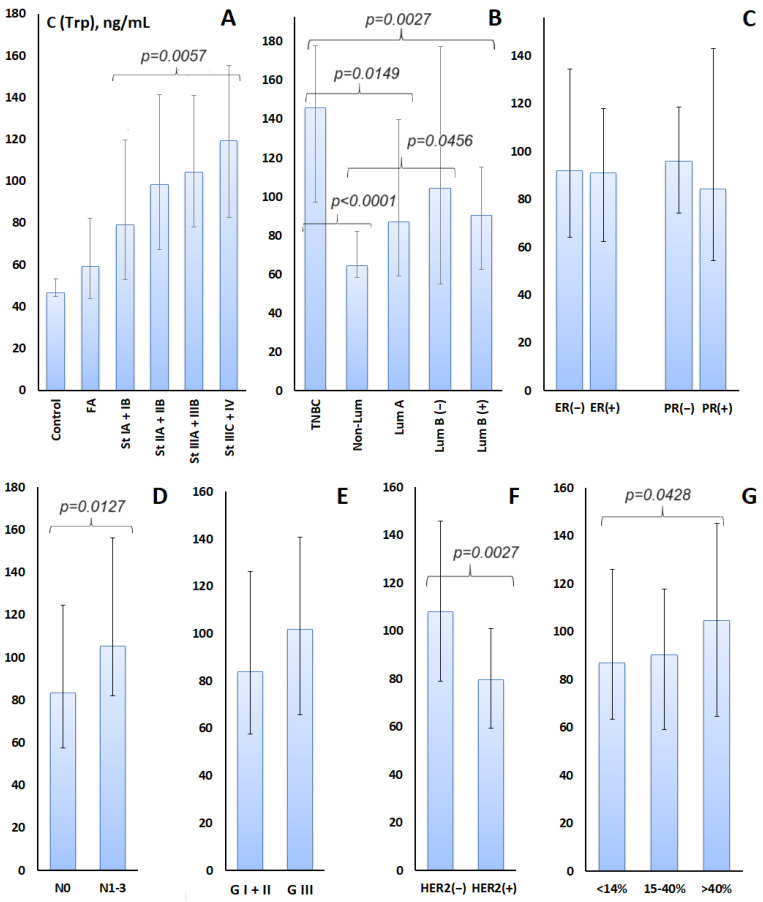
Tryptophan concentration in the studied subgroups: (**A**)—control group, fibroadenomas and breast cancer, depending on the stage of breast cancer; (**B**)—depending on the molecular biological subtypes of breast cancer; (**C**)—depending on the expression of estrogen and progesterone receptors; (**D**)—depending on the damage to the lymph nodes; (**E**)—depending on the degree of tumor differentiation; (**F**)—depending on HER2-positive and -negative breast cancer status; (**G**)—depending on the expression of the proliferation marker Ki-67. For groups whose differences are statistically significant, *p*-values are given (*p* < 0.05).

**Figure 2 metabolites-14-00247-f002:**
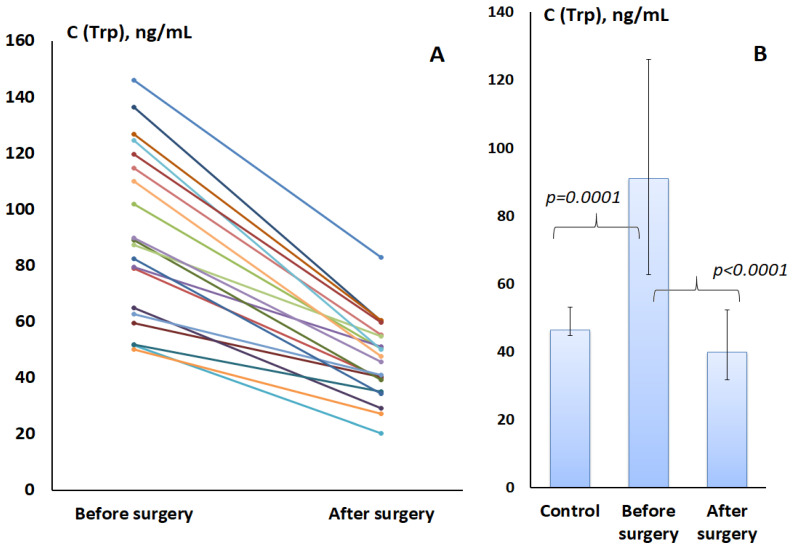
Changes in the concentration of tryptophan in saliva before and after surgery for 20 patients with breast cancer: (**A**)—for individual patients with an interval of 4 weeks; (**B**)—in comparison with the tryptophan content in the control group.

**Figure 3 metabolites-14-00247-f003:**
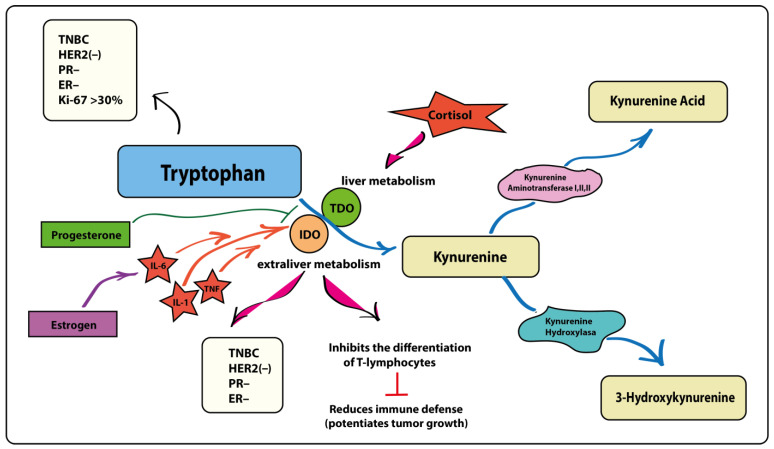
TNF—Tumor necrosis factor-α; IL-1—interleukin-1β; IL-6—interleukin-6; IDO—indoleamine 2,3-dioxygenase; TDO—tryptophan-2,3-dioxygenase; TNBC—triple negative breast cancer; HER2(−)—human epidermal growth factor receptor 2 negative; PR(−)—progesterone receptor negative; ER(−)—estrogen receptor negative; Ki-67—marker of proliferation Ki-67.

**Table 1 metabolites-14-00247-t001:** Trp content in saliva depending on the characteristics of breast cancer.

Feature		Breast Cancer, n = 103	Trp, µg/L	*p*-Value
**Clinical Stage**				
	Stage IA + IB	33 (32.0%)	79.09 [52.74; 139.7]	0.1427
	Stage IIA + 2B	40 (38.8%)	98.33 [67.10; 141.4]	0.0055 *
	Stage IIIA + IIIB	15 (14.6%)	104.3 [78.06; 140.9]	0.0284
	Stage IIIC + IV	15 (14.6%)	119.1 [82.48; 175.3]	0.0091
**Lymph node status**				
	N_0_	58 (56.3%)	83.50 [57.62; 124.7]	0.0660
	N_1-3_	45 (43.7%)	105.2 [82.01; 176.3]	0.0004
**Subtype**				
	Luminal A-like	22 (21.4%)	87.01 [58.96; 139.7]	0.0289
	Luminal B-like (HER2+)	21 (20.4%)	90.58 [62.39; 115.3]	0.1158
	Luminal B-like (HER2−)	24 (23.3%)	104.3 [55.03; 177.3]	0.0262
	HER2-enriched (Non-Lum)	16 (15.5%)	64.57 [58.50; 81.97]	0.6411
	Triple-negative	20 (19.4%)	145.96 [97.18; 177.7]	0.0001
**HER2 status**				
	HER2-negative	66 (64.1%)	108.0 [78.98; 166.0]	0.0005
	HER2-positive	37 (35.9%)	79.43 [59.39; 101.9]	0.2014
**Estrogen (ER) status**				
	ER-negative	38 (36.9%)	91.86 [64.20; 154.6]	0.0080
	ER-positive	65 (63.1%)	90.98 [62.39; 137.9]	0.0089
**Progesterone (PR) status**				
	PR-negative	68 (66.0%)	95.93 [74.03; 138.6]	0.0064
	PR-positive	35 (34.0%)	84.08 [54.18; 162.8]	0.0599
**Degree of differentiation (G)**				
	G I + II	51 (49.5%)	83.92 [57.62; 146.1]	0.0203
	G III	39 (37.8%)	101.7 [65.50; 140.9]	0.0083
**Ki-67**				
	<14%	27 (26.2%)	87.01 [63.46; 146.1]	0.0127
	15–40%	36 (35.0%)	90.26 [59.17; 117.9]	0.0627
	>40%	40 (38.8%)	104.7 [64.57; 165.3]	0.0037

Note: *—Differences with the fibroadenomas group are statistically significant at *p* < 0.05.

## Data Availability

The raw data supporting the conclusions of this article will be made available by the authors on request.

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
