# Peer review of "Salivary Tryptophan as a Metabolic Marker of HER2-Negative Molecular Subtypes of Breast Cancer"

_metabolites, 2024, doi:10.3390/metabo14050247_

Round 1

Reviewer 1 Report

Comments and Suggestions for Authors

The study examined changes in salivary tryptophan (Trp) concentrations in patients with breast cancer, considering it as indicative of serious metabolic restructuring related to various diseases, and concludes that salivary Trp determination could be a valuable tool in studying metabolic changes associated with breast cancer. This study is interesting and valuable. However, the data presentation is unclear and unscientific, and part of the discussion is not relevant to the content.

Major criticizes:

In lines 101-103, the author indicated in Table 1 that the p-values are given in comparison with fibroadenomas and noted that the differences with the control group are statistically significant in all cases. Therefore, Figure 1, which presents the same results, is unnecessary and redundant. Additionally, the method used to denote the statistical differences is unscientific. It would be more meaningful to display the statistical differences among subgroups, as described in lines 121-127, and to use a scientific approach to denote significance, such as * for p<0.05 and ** for p<0.01.

Line 131: Is the difference significant here, and what is the p-value? The same questions should be answered for lines 132 to 138 and line 143.

Line 147-150: Line 147-150: The text description is unclear and confusing. Please use a table to help and list the p-values for all significant comparisons.

Line 231-259: The discussion part is irrelevant to the content.

Comments on the Quality of English Language

1)      Line 22: Consider adding "the" before "high proliferative activity" for consistency.

2)      Line 99: “the Trp content increases slightly – 59.19 [43.85; 82.01] μg/L (p=0.0960).” should be “the Trp content increases slightly to 59.19 (43.85; 82.01) μg/L (p=0.0960).”

3)      Line 100: Consider rephrasing "statistically significantly increases" to "increases significantly" for smoother flow. Use "compared with" instead of "compared to" for better clarity. Revised text: In breast cancer, the Trp content increases significantly to 90.98 (62.68; 146.1) μg/L compared with both the control group (p<0.0001) and fibroadenomas (p=0.0039).

Reviewer 2 Report

Comments and Suggestions for Authors

Authors reported that the levels of tryptophan (Trp) in the saliva of breast cancer patients, non-breast cancer and healthy controls. The concentration of Trp was high in breast cancer patients which was significantly decreased after the surgical removal of the tumors. Further, a positive correlation was found between the levels of Trp and the degree of malignancy, aggressive subtype of HER2 negative breast cancer. Overall, this study was performed very well by evaluating the levels of Trp in the clinical samples of different breast cancer subtypes. However, this study has many potential limitations, but it is an important study as the salivary Trp levels used for detection of metabolic changes associated with HER2 negative breast cancer subtype.

There is always room to ask many questions but, in each study, it is important to see the novelty and how it is relevant for the field. For example,

1.     Authors should evaluate other metabolites such as IDO1, TDO2 and Kynurenine to understand the role of Trp catabolism in breast cancer patients.

2.     It is important to study the correlation between tryptophan and other metabolites (IDO1, TDO2 and Kynurenine).

Therefore, I recommend this manuscript for publication after addressing above issues.

Comments on the Quality of English Language

Minor language editing is required

Round 2

Reviewer 1 Report

Comments and Suggestions for Authors

Thank you for the corrections. They have significantly improved the manuscript.

Comments on the Quality of English Language

No comments.